# Early cellular innate immune responses drive Zika viral persistence and tissue tropism in pigtail macaques

Megan A. O'Connor [1,2], Jennifer Tisoncik-Go[3,4], Thomas B. Lewis[1,2], Charlene J. Miller[5,7], Debra Bratt [2], Cassie R. Moats[2,8], Paul T. Edlefsen [6], Jeremy Smedley [2,8], Nichole R. Klatt[2,5,7], Michael Gale Jr[3,4] & Deborah Heydenburg Fuller[1,2]

The immunological and virological events that contribute to the establishment of Zika virus (ZIKV) infection in humans are unclear. Here, we show that robust cellular innate immune responses arising early in the blood and tissues in response to ZIKV infection are significantly stronger in males and correlate with increased viral persistence. In particular, early peripheral blood recruitment of plasmacytoid dendritic cells and higher production of monocyte chemoattractant protein (MCP-1) correspond with greater viral persistence and tissue dissemination. We also identify non-classical monocytes as primary in vivo targets of ZIKV infection in the blood and peripheral lymph node. These results demonstrate the potential differences in ZIKV pathogenesis between males and females and a key role for early cellular innate immune responses in the blood in viral dissemination and ZIKV pathogenesis.

[1] Department of Microbiology, University of Washington, Seattle, 98195 WA, USA. [2] Washington National Primate Research Center, Seattle, 98121 WA, USA. [3] Department of Immunology, University of Washington, Seattle, 98109 WA, USA. [4] Center for Innate Immunity and Immune Disease (CIIID), University of Washington, Seattle, 98109 WA, USA. [5] Department of Pharmaceutics, University of Washington, Seattle, 98195 WA, USA. [6] Vaccine and Infectious Disease Division, Fred Hutchinson Cancer Research Center, Seattle, 98109 WA, USA. [7] Present address: Department of Pediatrics, University of Miami, Miami, 33136 FL, USA. [8] Present address: Oregon National Primate Research Center, Hillsboro, 97006 OR, USA. Correspondence and requests for materials should be addressed to D.H.F. (email: fullerdh@wanprc.org)

Zika virus (ZIKV) was first isolated in 1947 from a sentinel rhesus macaque in the Zika forest and is a global epidemic in recent years[1]. The primary means of ZIKV transmission is via a mosquito bite; however, transmission also occurs through sexual intercourse, from mother to fetus, and potentially through other bodily fluids[2,3]. ZIKV infection results in mild and usually self-limiting symptoms including rash, fever, and conjunctivitis, but is associated with Guillain–Barré syndrome (GBS) in adults, and ZIKV exposure during fetal development can result in abnormalities in the fetus and infant[4,5]. Clinical diagnosis in the early course of ZIKV infection has been difficult to discern from other flaviviruses such as dengue (DENV) and West Nile (WNV) viruses because clinical symptoms overlap and ZIKV occurs in the same endemic areas as these viruses.

ZIKV infection in nonhuman primate (NHP) models including rhesus, pigtail, and cynomolgus macaques is mild and self-limiting, and the disease course is similar to human infection, therefore NHPs provide valuable models for studying the disease[6–13]. Vertical ZIKV transmission also occurs in NHPs, resulting in prolonged viremia in the mother and detection of ZIKV and developmental abnormalities in the fetus that closely resembles ZIKV pathogenesis in humans, including fetal ocular pathologies and brain injury[6,10,14–17]. The NHP is also a highly relevant model to evaluate ZIKV pathogenesis by various means of exposure that occur in humans[6,13,14,18,19] and for testing the efficacy of ZIKV vaccines and therapeutics[20,21]. In symptomatic human cases, clinical symptoms do not arise until 3–11 days post-exposure[22], limiting the ability to study the earliest stages of ZIKV infection in humans. In contrast, due to the ability to control the timing of the infection in NHPs, ZIKV can be investigated at the earliest stages following exposure in vivo, including early immunological and virological events and their role in pathogenesis.

To date, studies in NHP models have shown an important role for innate immune cells in ZIKV infection. Recruitment and activation of monocytes, dendritic cells, neutrophils, and/or NK cells into the blood, and induction of pro-inflammatory cytokines and signaling pathways occur during ZIKV infection in rhesus macaques (RM)[6,7,9,23,24]. In RMs, ZIKV persists longer in lymph nodes and persistence has been linked with the activation of mTOR, a target of rapamycin, pro-inflammatory, and anti-apoptotic signaling pathways[24]. In addition, studies in RM have shown that innate immune cells, including those of myeloid and neutrophil origin[7,9], can be infected with ZIKV, suggesting a potential role for these innate immune cells in ZIKV pathogenesis. However, the analysis of innate immune cell dynamics in response to infection in humans or NHPs has mostly been limited to blood and plasma. Since ZIKV has been detected in a broad range of tissues including lymphoid, brain, urogenital, and mucosal tissues[7,8], we employed a pigtail macaque (PTM) model of ZIKV infection to investigate the innate immune cell response in several tissues including blood, lymph nodes, and mucosa from the earliest stages of infection and their potential role in ZIKV pathogenesis.

Here, we show that ZIKV infection induces a robust cellular innate immune response including rapid dendritic cell, monocyte, and neutrophil recruitment into the blood and tissues within the first week of infection and, interestingly, greater ZIKV persistence and broader tissue tropism, especially in the males, correlates with higher frequencies of plasmacytoid dendritic cells (pDCs) in the blood and higher levels of MCP-1 production in the plasma within the first 4 days of infection. Furthermore, we demonstrate that non-classical monocytes are primary cellular targets for ZIKV infection in vivo and may drive ZIKV viral dissemination and persistence. These results provide insights into the specific cellular innate immune responses arising at the earliest stages post-ZIKV exposure and potentially a central role for these responses in ZIKV infection, persistence, and pathogenesis. Additionally, the findings in these studies provide a greater understanding of the early cellular events that may influence ZIKV infection in humans.

## Results

**Experimental timeline and ZIKV kinetics**. Eight pigtail macaques (PTM) (Cohort 1: four non-pregnant females, Cohort 2: four males) were subcutaneously infected into the forearm with $5 \times 10^5$ PFU of ZIKV (Brazil_2015_MG, GenBank: KX811222.1. All animals were pre-screened for the presence of antibodies to WNV, DENV, and ZIKV, and all were negative except for a single female animal (L07226), who was seropositive for West Nile virus. Blood was collected daily during the first 4 days post-infection (dpi) and then every 2–3 days starting at 7 dpi (Fig. 1a). Tissues (lymph nodes and mucosal tissues) were collected prior to infection and at 7, 14, and 21 dpi (Fig. 1a), and viral burdens were measured at each specimen collection timepoint by RT-qPCR. Plasma viremia was detected on day 1, peaked 2–3 dpi, and was completely resolved in all animals by 7 dpi (Fig. 1b). No differences in plasma viral kinetics or burden were observed between the males and females (Fig. 1b–d), and plasma viremia in the PTMs studied here closely resembles viremia previously reported in RMs infected with similar strains of Asian-lineage ZIKV[6–9,11,23,24]. One of the female animals (L07266) was WNV seropositive prior to ZIKV infection; however, this animal did not appear to have enhanced viremia, which supports the previous studies in the RM demonstrating a lack of enhancement of ZIKV infection in flavivirus-immune animals[25,26]. L07266 cleared the virus in the peripheral lymph node (PLN) by day 7, more quickly than the other animals (Fig. 1b, c). However, pre-infection anti-ZIKV envelope IgG or IgA antibodies were not higher in this animal, and baseline levels did not correspond to lower ZIKV viral burden in the PLN of L07266 when compared with WNV-negative animals, indicating pre-existing immunity to WNV did not contribute to the early viral clearance in the PLN in this animal (Supplementary Fig. 1). Although the virus was undetectable in the plasma by 7 dpi, ZIKV could still be detected at 21 dpi in the PLN in six of the eight animals (Fig. 1b, c). This finding is consistent with the previous reports of RM, where ZIKV was similarly found to persist longer in the lymph node[7,24]. Interestingly, we observed greater viral persistence and burden in the PLN of males as compared with females (Fig. 1b–d). To determine if the virus also persisted in the mucosal tissues, we additionally measured the virus in the jejunum, colon, rectum, and vagina at 7, 14, and 21 dpi. At 7 dpi, ZIKV was detected in the rectum of three males, one of the males (K11095) also had ZIKV in the colon at 7 dpi (Fig. 1b, c). Although ZIKV RNA was detectable in the rectum and colon at 7 dpi, due to limited number of biopsies collected, we were unable to determine if the virus in these tissues was infectious. In contrast, virus was undetectable in the mucosa (jejunum, colon, rectum, or vaginal tissues) in any of the females at all timepoints tested (Fig. 1b–d). Collectively, these results show that the males exhibited more viral persistence in the PLN and viral dissemination to tissues (rectum, colon) than the females (Fig. 1b–d).

**Innate immune cell responses after ZIKV infection**. Innate immune cell responses were evaluated in the blood and tissues throughout ZIKV infection. To account for differences in pre-infection immune frequencies between the sexes that could influence the kinetics and magnitude of the host response to ZIKV infection, we evaluated the area under the curve (AUC) of

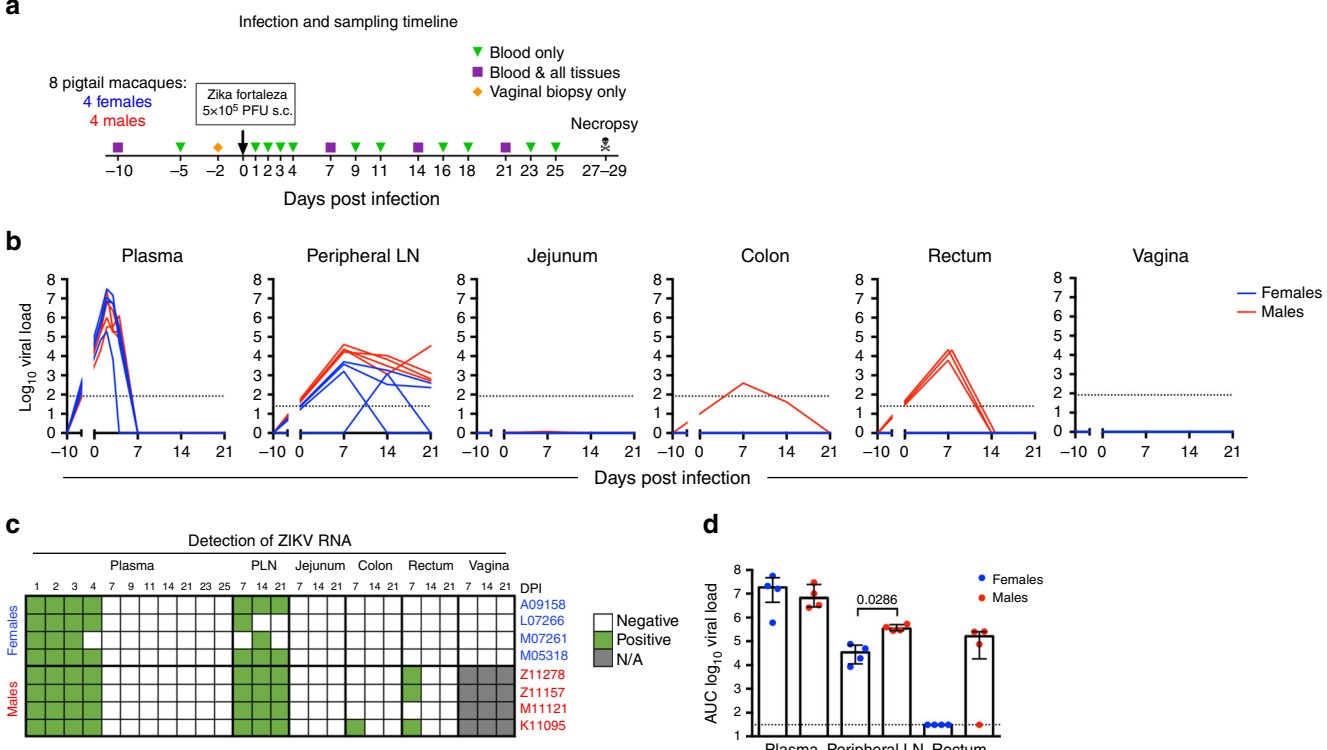

**Fig. 1** Viral persistence in LNs and viral dissemination to mucosal tissues are greater in male PTMs. **a** Eight pigtail macaques (n = 4 females, n = 4 males) were subcutaneously (s.c.) infected with 5 × 10⁵ plaque-forming units (PFU) of a Brazilian Fortaleza isolate of ZIKV. Blood was routinely collected at 10 and 5 days prior to infection (−10 and −5 dpi), daily for the first 4 days post-infection, and then every 2–3 days until necropsy at 27–29 dpi. In addition to blood draws, comprehensive biopsy samples of the gastrointestinal tract (jejunum, colon, and rectum) and whole lymph nodes (peripheral) at −10, 7, 14, and 21 dpi were taken. Vaginal biopsies were also sampled 2 days prior to infection (−2 dpi) and 7, 14, and at 21 dpi. **b** Kinetics of ZIKV RNA levels in the plasma (Log₁₀ copies/ml) and various tissues (Log₁₀ copies/mg) were determined by RT-qPCR. Dotted line indicates the limit of detection: 83 copies/ml or mg in plasma, jejunum, colon, and vagina or 25 copies/mg in the peripheral lymph node and rectum. **c** ZIKV distribution in plasma, peripheral lymph node (PLN), and mucosal tissues at 0–25 days post-infection (DPI) (plasma) or 0–21 dpi (PLN, mucosal tissues) is shown as follows: ZIKV RNA positive (green), negative (white), or tissue not available (N/A) (gray). **d** ZIKV burdens were determined by measuring area under the curve (AUC) in plasma (0–25 dpi), PLN, and rectum (0–21 dpi) in female (blue) and male (red) animals. Dotted lines indicate the limit of detection (31 copies/ml or mg) of the AUC. Dots are individual animals (total n = 8) and bars are the medians with interquartile ranges. Significant (unadjusted Wilcoxon p ≤ 0.05) p-values are displayed

the immune cell responses. AUC analysis was between days −10 to +4 in the blood and between days −10 to +21 in the tissues. We first examined dendritic cell populations including plasmacytoid DCs (pDCs; CD14⁻CD123⁺CD11c⁻), a known type I interferon (IFN)-producing cell type, and myeloid DCs (mDCs; CD14⁻CD123⁻CD11c⁺), an antigen-presenting cell that can prime adaptive immune responses. In males, but not in females, pDCs were the first cells to respond to the infection in the blood, with substantial and sustained recruitment at 1–4 dpi (Fig. 2a, Supplementary Table 1, 2). In the male cohort, blood pDCs also exhibited prolonged increases in CD86 expression, a marker of DC activation, during the first week of infection (Supplementary Fig. 2). Changes in mDC frequencies in the blood were not detected in any of the animals (Fig. 2b), but a marked decrease in CD86 expression on these cells occurred in the first 2 days, with recovery to baseline levels by 4 dpi (Supplementary Fig. 2, Supplementary Table 3). Increased frequencies of pDCs and mDCs were also detected in mucosal sites including the rectum (Fig. 2a, b) and in the jejunum and colon (Supplementary Fig. 3, Supplementary Table 2); at the first timepoint, these tissues were collected (7 dpi) and remained elevated in several animals up to 21 dpi. The pDC and mDC responses were less robust in the PLN (Fig. 2a, b). In summary, pDCs were early innate responders to ZIKV infection in the blood, and DCs (pDCs and mDCs) were

recruited to mucosal sites during the first 21 days post-ZIKV exposure.

The influx of monocytes (HLA-DR⁺CD14±CD16±) post-infection was also assessed and included classical (CD14⁺CD16⁻) and intermediate (CD14⁺CD16⁺) subsets that produce inflammatory cytokines and non-classical monocytes (CD14intCD16⁺) that can have an anti-viral role including viral RNA detection by toll-like receptors. In all animals, blood monocytosis occurred 2–4 dpi (Fig. 2c), a finding that is consistent with studies in ZIKV infected patients and other NHP models[9,27,28], corresponding to increases in all monocytes subsets within the first 4 days of infection (Supplementary Fig. 4) and declines in CD86 expression in classical blood monocytes occurred throughout ZIKV infection starting at 1 dpi and extending through 25 dpi (Supplementary Fig. 2). Similar declines in CD86 expression were also observed on intermediate and non-classical monocytes (Supplementary Fig. 2). In the mucosal tissues, significant monocyte recruitment occurred in females, but not in males (rectum, jejunum, and colon, all p = 0.0286, Wilcoxon test) (Fig. 2c, Supplementary Fig. 3, Supplementary Table 1), with variable recruitment of monocyte subsets depending on the tissue (Supplementary Fig. 4). In all animals, monocyte decline occurred 7–21 dpi in the PLN (Fig. 2c), with a notable decrease in classical monocytes at 7–21 dpi (Supplementary Fig. 4). In the lymph node and mucosal tissues, we identified

monocytes using the same gating scheme as in the blood, which excluded dendritic cells, but may include monocyte-derived and macrophage cell populations. By 14 dpi, all females showed a shift toward higher proportions of CD16$^-$ classical monocytes in the jejunum, colon, rectum, and vagina (Supplementary Fig. 4). CD16$^+$ monocytes have been implicated as cellular targets of ZIKV infection in the blood[27,28], but our findings also showed the CD16$^-$ monocyte recruitment into mucosal tissues in the females in the absence of detectable ZIKV RNA in these tissues (Fig. 1b, Supplementary Fig. 4). These results may be due to having a limited amount of tissue to analyze viral loads, but also suggest that monocytes migrating to these tissues could have an anti-viral role.

Neutrophils have been shown to be highly responsive to flavivirus infection. During dengue infection neutrophils are depleted in the blood, and WNV meningitis and encephalitis are associated with neutrophilia in the cerebral spinal fluid (CSF)[29]. In ZIKV-infected animals studied here, an initial, but insignificant decline in blood neutrophils (CD11b$^+$CD14$^+$SSC-A$^{hi}$) occurred in the female blood at 2 dpi followed by a delayed increase in neutrophils at 9 dpi in all animals (Fig. 2d). In the tissues, neutrophil recruitment to the vagina at 7 dpi and significant neutrophil influx to the female gut tissues at 7–21 dpi including the rectum (Fig. 2d), jejunum, and colon (Supplementary Fig. 3) also occurred. Initial baseline levels of neutrophils in the blood and mucosal tissues were also higher in the females, which may be a factor that contributed to greater recruitment of these cells in the females after ZIKV infection. In the PLN, low neutrophil frequencies were detected in all animals with no significant changes after ZIKV infection (Fig. 2d). Together these results show, especially in the females, neutrophil recruitment to mucosal sites during acute infection and delayed neutrophil recruitment to blood in all animals following viral clearance.

**Induction of pro-inflammatory cytokines after ZIKV infection.** Several types of innate immune cell subsets that responded to ZIKV infection in our study, including monocytes and dendritic cells, are known to be potent immunomodulators during flavivirus infections. To determine if increases in these cell subsets corresponded to the induction of pro-inflammatory mediators, we measured the effects of ZIKV infection on the levels of 23 plasma cytokines and chemokines prior to infection, and at peak plasma viremia (2 dpi, Fig. 3) by Luminex xMAP multiplex. Overall, we observed significant increases in several pro-inflammatory cytokines/chemokines only at 2 dpi, including interleukin 1 receptor antagonist (IL-1RA), an inhibitor of IL-1α and IL-1β, soluble CD40 ligand (sCD40L), which can trigger the release of inflammatory mediators, the monocyte chemoattractant protein (MCP-1 (CCL-2)), and interleukin 15 (IL-15), which regulates T- and NK-cell proliferation and activation, respectively (Fig. 3a). We also observed sex differences in certain responses. In particular, males had higher MCP-1 levels at 2 dpi (Fig. 3b, $p = 0.0286$, Wilcoxon test), and plasma MCP-1 levels positively correlated with higher levels of pDCs in the blood at this timepoint (Supplementary Fig. 5, $p = 0.0158$, Spearman's rank). In contrast, neutrophil chemoattractant interleukin 8 (IL-8) was significantly higher in female plasma at 2 dpi (Fig. 3b, $p = 0.0286$, Wilcoxon test), and higher plasma IL-8 positively correlated with increased neutrophil recruitment to the blood at 9 dpi (Supplementary Fig. 5, $p = 0.0154$, Spearman's rank). Females also tended to have higher plasma levels of sCD40L ($p = 0.0571$, Wilcoxon test) compared with males (Fig. 3b), but these parameters did not significantly correlate with peak innate immune cell subsets in the blood during the early phase of infection. Together, these results are consistent with studies

showing induction of pro-inflammatory cytokines following ZIKV infection[7,9] and demonstrate that early production of cytokines and chemokines in the plasma correspond to subsequent recruitment or increases in the frequency of innate immune cells to the blood.

**In vivo targets of Zika infection.** We next investigated if innate immune cell subsets responding to the infection might also contribute to ZIKV pathogenesis by serving as early targets of infection. To identify cell subsets infected with ZIKV, whole blood (1–3 dpi) and cells isolated from the PLN (7–14 dpi) were stained intracellularly for ZIKV non-structural 3 (NS3) protein using an antibody specific for WNV NS3 (Fig. 4). During peak viremia measured at 1–3 dpi, NS3 expression was primarily found in mDCs and non-classical monocytes (CD14$^{int}$CD16$^+$) in the blood (Fig. 4a), suggesting that these cell types may be major in vivo blood targets of infection. Consistent with this possibility, we found that among all animals, higher frequencies of mDCs ($p = 0.0063$ Spearman's rank) and non-classical monocytes ($p = 0.0047$, Spearman's rank) present in the blood within the first 3 days of infection significantly correlated with increased levels of ZIKV plasma viremia during this timeframe (Fig. 4b). Similar to the findings in the blood, NS3 staining in the PLN at 7–14 dpi showed an expression primarily in non-classical monocytes (Fig. 4c). In addition, NK cells in the PLN also expressed higher levels of NS3 (Fig. 4c), indicating that NK cells localized in the lymph nodes may also serve as early in vivo targets of ZIKV infection. Our analysis of NK-cell frequency over time showed no significant increase in this subset following ZIKV infection in the blood or tissues tested, suggesting that NK cells, while an early target of ZIKV infection, may have limited involvement in ZIKV pathogenesis (Supplementary Fig. 6). The frequencies of mDCs ($p = 0.0857$, Spearman's rank) and non-classical monocytes ($p = 0.0821$, Spearman's rank) (Fig. 4d), but not NK cells ($p = 0.4909$, Spearman's rank) (Supplementary Fig. 7), trended toward correlating with the viral burden in the PLN; although possibly due to the small sample group, these comparisons fell short of statistical significance. Collectively, these results demonstrate that non-classical monocytes are major early targets of ZIKV infection in the blood and lymph nodes.

**Innate responses contribute to viral persistence and tropism.** Figure 1b–d shows that the PLN is a site of ZIKV persistence and the rectum is a site of significant viral tropism in males. To determine if there is a relationship between the early cellular innate immune response in the blood to ZIKV infection and viral persistence or tissue tropism, we next examined the correlations between specific innate responses measured in the PLN and rectum and the level of ZIKV burden in these tissues. Cellular innate immune responses measured in the rectum did not correlate with ZIKV RNA levels in the rectum at 7 dpi (Supplementary Table 4), suggesting that local innate responses may not contribute to viral replication in this tissue at this timepoint. Analysis of the frequencies of mDCs and non-classical monocytes in the PLN showed a trend ($p = 0.08$, Spearman's rank) toward correlating with higher levels of viral loads in this tissue, but fell short of statistical significance (Fig. 4d). We next examined if early cellular innate immune responses measured prior to infection at −10 to +4 dpi in the blood correlated with viral burden in these two tissues. Only pDC frequencies in the blood, measured as AUC prior to infection at −10 to +4 dpi, correlated with AUC levels of ZIKV RNA measured in the PLN ($p = 0.0046$, Spearman's rank) during the first 21 days of infection (including pre-infection) (Fig. 5b) and in the rectum at day 7 ($p = 0.0357$, Wilcoxon test) (Fig. 5a).

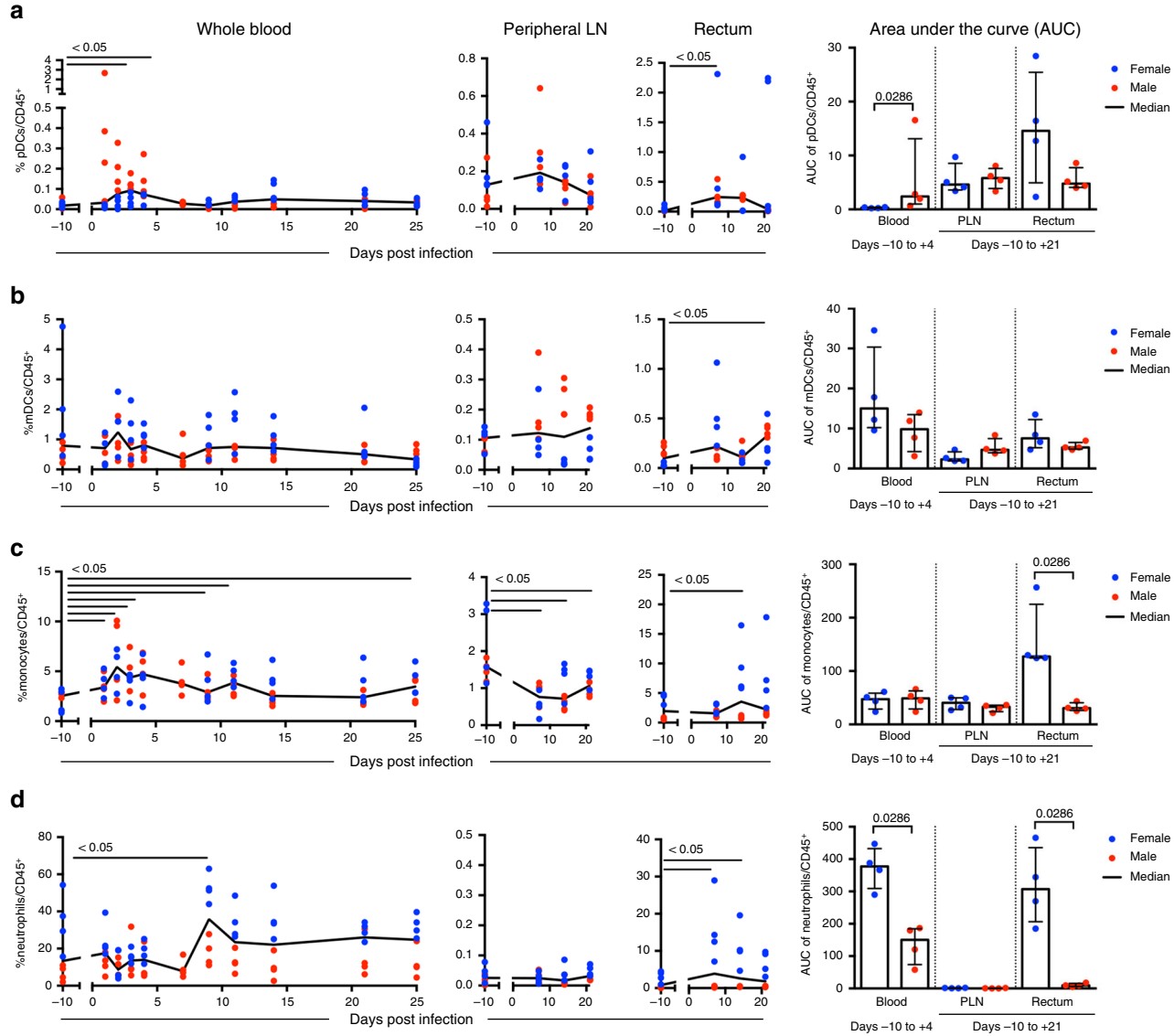

**Fig. 2** Innate immune cells are rapidly recruited in response to ZIKV infection. Frequencies within CD45$^+$ leukocytes and area under the curve (AUC) analysis of **a** pDCs (CD20$^-$CD3$^-$HLA$^-$DR$^+$CD14$^-$CD123$^+$CD11c$^-$), **b** mDCs (CD20$^-$CD3$^-$HLA$^-$DR$^+$CD14$^-$CD123$^-$CD11c$^+$), **c** monocytes (CD20$^-$CD3$^-$HLA$^-$DR$^+$CD16$^\pm$CD14$^\pm$), and **d** neutrophils (CD3$^-$CD11b$^+$CD14$^+$HLA-DR$^-$SSC-A$^{Hi}$) in whole blood, peripheral lymph node (PLN), and rectum throughout the course of ZIKV infection were measured by flow cytometry. The frequency of these responses and AUC analysis in blood (AUC, days −10 to +4) and tissues (AUC, days −10 to +21) are shown. **a–d** Blue (females) and red (males) symbols in graphs represent individual animals, with curves indicating the median response for all animals over time. Comparisons of responses at each timepoint versus baseline were conducted by paired Wilcoxon test. Unadjusted p-values ≤ 0.05 are displayed, all p-values are available in Supplementary Table 2. **a–d** Differences in AUC between males and females were also determined using the Wilcoxon test, with significant unadjusted p-values (p ≤ 0.05) shown. Dots represent individual animals (total n = 8), female (blue) and male (red), with bars indicating median with interquartile ranges

To determine if innate responses contributed to the observed differences in viral persistence and tropism in the tissues of the males (Fig. 1b–d), we next focused on analyzing the correlations between innate responses, measured by Luminex, that differed markedly between males and females (plasma sCD40L, MCP-1, and IL-8 (Fig. 3b)) to levels of ZIKV RNA in the PLN and rectum. Among these, only MCP-1 production measured at 2 dpi significantly correlated with levels of ZIKV RNA measured in the PLN at −10 to +21 dpi (p = 0.0046, Spearman's rank) (Fig. 5b) and rectum at 7 dpi (p = 0.0357, Wilcoxon test) (Fig. 5a). MCP-1 is a chemoattractant for monocytes and dendritic cells, suggesting that MCP-1 expression may play a role in the recruitment of monocytes or dendritic cells we observed following ZIKV infection. Consistent with this possibility, higher plasma levels of MCP-1 at 2 dpi correlated with increased recruitment of mDCs (p = 0.0218, Spearman's rank) and non-classical monocytes (p = 0.0279, Spearman's rank) (Fig. 5c) to the PLN. Collectively, these data suggest an important and early role for pDCs and the expression of MCP-1 in ZIKV pathogenesis. Specifically, our data provide evidence that following ZIKV infection, pDC recruitment and the production of MCP-1 may initiate the recruitment of non-classical monocytes and other ZIKV cellular targets to the lymph nodes, resulting in the observed increased in ZIKV burden and persistence at this site.

## Discussion

Using the pigtail macaque (PTM) model of ZIKV infection, we demonstrate rapid development of cellular innate immune

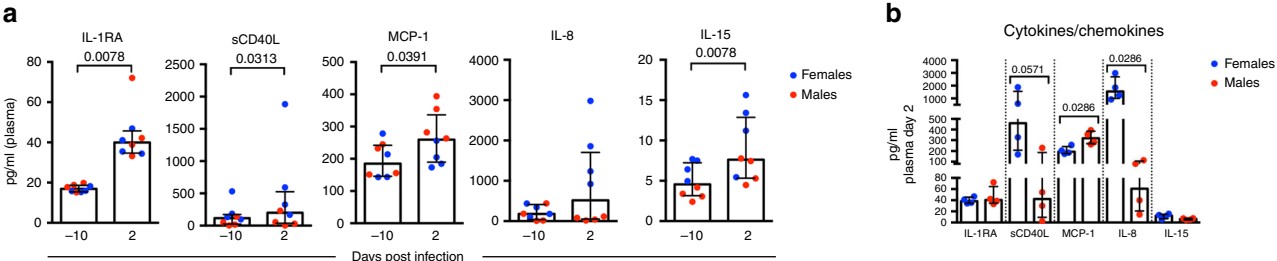

**Fig. 3** Induction of pro-inflammatory cytokines following ZIKV infection. **a** Concentrations (pg ml$^{-1}$) of plasma IL-1RA, sCD40L, MCP-1, IL-8, and IL-15 at baseline (−10 dpi) and 2 dpi were measured by Luminex. Blue (females) and red (males) symbols indicate individual animals ($n = 8$). Medians are indicated by bars with interquartile ranges. Differences between responses at baseline and 2 dpi were evaluated using paired Wilcoxon tests ($n = 8$), with unadjusted $p$-values displayed and significant differences defined as $p \leq 0.05$. **b** Concentrations (pg ml$^{-1}$) of plasma cytokines and chemokines at 2 dpi were measured by Luminex. Symbols indicate individual animals ($n = 8$), females (blue) or males (red). Medians are indicated by bars and show interquartile ranges. Differences between males and females were evaluated using Wilcoxon tests ($n = 4$ per group), and the corresponding unadjusted significant ($p \leq 0.05$) or trending ($p \leq 0.08$) $p$-values are shown

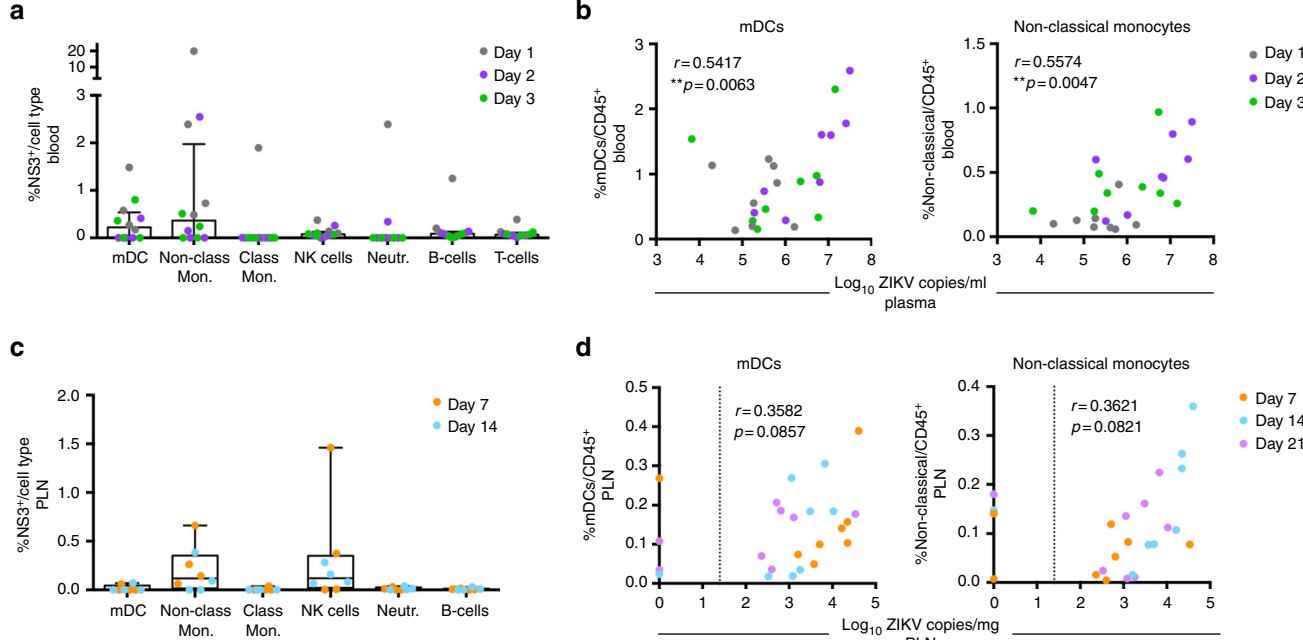

**Fig. 4** Non-classical monocytes are the major in vivo cellular target of ZIKV infection. **a**, **b** ZIKV-infected cells isolated from blood or **c**, **d** PLN in the males were identified by flow cytometry following intracellular staining for ZIKV-NS3 and cell surface staining to identify immune cell subsets ($n = 4$/timepoint). Positive cells were identified after background subtraction and adjusting for autofluorescence. The threshold limit was ≥100 cells/gate, and cell types that did not reach this threshold for all animals at all timepoints are excluded. **a**, **c** Percentages of NS3$^+$ cells within myeloid DCs (mDCs), non-classical monocytes (Non-Class. Mon.), classical monocytes (Class. Mon.), NK cells, neutrophils (Neutr.), B cells, or T cells are shown 1, 2, and 3 dpi in blood and 7 and 14 dpi in the PLN. Dots are individual animals (males) with box plots showing the median with interquartile ranges. **b** Scatter plot and Spearman's correlation analysis of early (1–3 dpi) plasma viremia and frequencies of mDCs and non-classical monocytes in the blood at matched timepoints (1–3 dpi). **d** Correlation between viral load in the PLN and frequencies of mDCs and non-classical monocytes in the PLN at matched timepoints (7–21 dpi). Dotted line indicates the limit of viral RNA detection (25 copies per mg). **b**, **d** Spearman's rank correlation coefficients are shown; significance of the unadjusted $p$-values are indicated as follows: **$p \leq 0.01$

responses in the blood and tissues in response to infection, and provide evidence for their role as targets for infection and in promoting viral tissue dissemination. Previously, transcriptomic analysis of peripheral blood mononuclear cells (PBMCs) from whole blood of rhesus macaques infected with ZIKV revealed a role for upregulated interferon responses, including increased IFI44, ISG15, MX2, and OAS1 gene expression, and a role for inflammatory monocytes positively correlating with plasma viral loads during acute infection[24]. Here, we further extend these findings and show that increases in blood pDCs and plasma MCP-1 in the first few days of infection are associated with

greater viral burden in the rectum and PLN and promote recruitment of non-classical monocytes to the lymph node (Fig. 5). Notably, we show that monocytes serve as early and primary cellular targets of in vivo ZIKV infection in the blood and lymph node and correlate with greater viral load at these sites (Fig. 4).

These results are consistent with other studies in the rhesus macaque that showed ZIKV infection of innate immune cells of myeloid and neutrophil origin in the spleen and lymph nodes[7,9], and studies in humans that showed ZIKV infection of monocytes in the blood[27,28], but extend these findings to identify that CD16$^+$

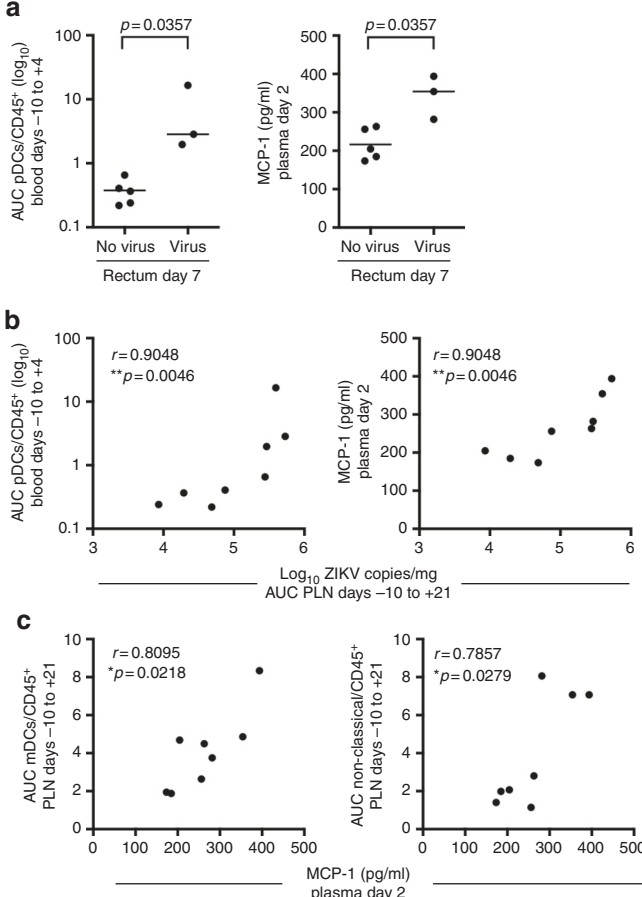

**Fig. 5** Innate responses in the blood recruit viral targets to tissues and promote viral persistence. **a** Early cellular innate immune responses in PTM with and without detectable ZIKV RNA in the rectum at day 7. Each panel shows the AUC of pDCs measured by flow cytometry in the blood between days −10 to +4 or the concentration of plasma MCP-1 measured at 2 dpi in animals with or without detectable ZIKV in the rectum at 7 dpi. Lines indicate the median. Differences between animals with no virus or virus in the rectum were evaluated using Wilcoxon tests. Unadjusted p-values are shown. **b** Correlations between viral burden in the PLN (AUC, days −10 to +21) and either pDC frequency (AUC, days −10 to +4) in the blood (left panel) or the concentration of plasma MCP-1 measured at 2 dpi (right panel). **c** Scatter plot and Spearman's correlation analysis of the relationship between the concentrations of plasma MCP-1 measured at 2 dpi and mDCs (AUC, days −10 to +21) and non-classical monocytes (AUC, days −10 to +21) in the PLN. Significance of the unadjusted Spearman's correlation test p-values are indicated as follows: *$p \leq 0.05$, **$p \leq 0.01$

monocytes are preferential early targets of ZIKV infection in both the blood and peripheral lymph nodes. These findings provide insight into the mechanisms of ZIKV pathogenesis and establish the PTM as a highly relevant model for evaluating ZIKV infection in humans. In humans, either non-classical (CD14− CD16+)[28] or intermediate (CD14+CD16+) monocytes[27] were identified as the main cellular blood targets. The disparate results in humans could be due to different ZIKV inoculums (H/PF/2013 versus Nica 2–16) or the study groups (adult versus children). However, overall results in humans are highly consistent with results reported here in the PTM model including preferential infection of CD16+ (intermediate, non-classical) monocytes by Asian-lineage ZIKV. Additionally, we identify that NK cells may also be targets of infection in the lymph node. In support of this

finding, CD16 is expressed on non-classical monocytes and NK cells, and infection of both of these cells types in PBMCs has been reported for ZIKV[27,28] and DENV[30]. The human cellular receptor for ZIKV entry is not currently known, however in vitro evidence suggests a role for dendritic cell-specific intercellular adhesion molecule 3-grabbing nonintegrin (DC-SIGN), found on both immature dendritic cells and macrophages, and/or the TAM receptors (AXL or Tyro3), expressed on monocytes, macro- phages, and NK cells[31,32]. These results support our finding that non-classical monocytes likely serve as early cellular targets of ZIKV infection.

As a permissive cell type to ZIKV, monocytes may assist with early viral dissemination and/or promote ZIKV persistence by creating a pro-inflammatory environment. In support of this, we found that higher levels of MCP-1 were associated with increased recruitment of non-classical monocytes to the PLN, a tissue we found to be a primary site of ZIKV persistence. Consistent with our findings, previous studies have shown that increased CCL2 expression and production of MCP-1 (CCL2) in the plasma occurs during human ZIKV or DENV infection[28,33], and CD16+ monocytes secrete MCP-1 in response to DENV infection[33,34]. Therefore, it is possible that non-classical monocytes also secrete MCP-1 in response to ZIKV infection. Our results suggest that in vivo ZIKV infection of CD16+ non-classical monocytes may lead to the production of inflammatory cytokines or chemokines, such as MCP-1, that may promote recruitment of more cellular targets and increased ZIKV dissemination and/or persistence. However, other cell types, including cells of the high endothelial venules[35], may also be a source of MCP-1 in the lymph nodes. Future studies in the PTM will be needed to determine the spe- cific immune responses that contribute to the recruitment of CD16+ non-classical monocytes and their role in ZIKV persis- tence and pathogenesis.

The host response and viral dynamics following ZIKV infec- tion in PTMs reported here, including a marked increase in monocyte frequency in the blood and increased production of pro-inflammatory cytokines/chemokines (IL-1RA, MCP-1, and IL-15) during peak viremia closely resembles responses to ZIKV infection previously reported in RMs[7,9], but differs in other measures including enhanced plasma sCD40L and IL-8 and lack of sporadic plasma viral blips following viral clearance at 7 dpi[6,7,9]. The use of different Asian-lineage ZIKV isolates in previous RM studies may account for some of the differences between the two species[6,7,9]. Notably, the increase in CD14+ CD16+ blood monocytes and plasma IL-8 observed here in the PTM model closely resemble the host response to ZIKV infection in humans[36,37]. In contrast to the RM that reproduces seasonally, the PTM also resembles humans in its year-round menstrual and reproductive cycles, making it a highly relevant model to study fetal complications induced by ZIKV infection[38]. Recent studies show that fetal brain malformations and the development of lesions in fetuses of infected pregnant PTMs also closely resemble the patterns in human congenital ZIKV infection[10]. The close similarity between the PTM and human response to ZIKV demonstrates the value of using this model to define mechanisms that contribute to human disease, and our results point to a key role for innate immune responses arising within the first few days after ZIKV exposure in viral tropism, persistence, and pathogenesis.

Notably, in this study, we observed significant differences in immunological and virological outcomes following ZIKV infec- tion in males versus females. Specifically, ZIKV persistence and burden in the PLN and rectum was greater in male PTMs and was associated with higher frequencies of pDCs in the blood and elevated plasma MCP-1 levels within the first 2 days of infection. It has been shown that pDCs are potent producers of type I

interferons[39]. Levels of circulating IFNα have been shown to increase in RMs within 24 h post-infection[25], but are not detected in ZIKV-infected marmosets post-ZIKV exposure[12]. Our cytokine analysis in response to ZIKV infection in PTMs did not include analysis of IFNα. Therefore, additional studies are needed in the PTM to determine if type I IFNs are produced in response to infection and if so, what is the cellular source of IFNs and whether type I IFN production contributes to viral clearance.

In females, we observed greater recruitment of CD16− monocytes and neutrophils to the rectum, gut, and vaginal mucosal tissues at 7–21 dpi, most likely in response to an acute ZIKV infection. Rapid immune cell trafficking to these tissues in females may contribute to earlier clearance of virus and account for the lack of detectable virus. The male and female cohorts were run separately and although we observed no differences in plasma viremia between the male and female cohorts, we cannot exclude the possibility that the observed sex differences in immune responses are due to temporal differences in the cohorts. In addition, other factors such age, weight, repeated ketamine sedation, and inherent differences in cellular frequencies prior to infection could alternatively influence cellular immune responses and associated sex differences[40]. However, sex differences in plasma viremia and immune responses have been previously reported in the RM[9], in agreement with the possibility of sex-disparate ZIKV pathogenesis and immune responses observed in our study. Sex differences in human ZIKV infection have also been reported, including higher rates of symptomatic cases in females, greater GBS rates in males, and in a few clinical studies, higher rates of fetal microcephaly in female infants[5,41–48]. Consistent with our findings showing greater viral persistence in the PLN of male PTMs, human studies have similarly shown that ZIKV persists longer in the semen, up to 3 months after infection, as compared with only transient infection, up to 2 weeks, in vaginal secretions[49–53]. The disparate virus localization and persistence in males may explain the higher male-to-female ZIKV sexual transmission rates that occur in the human population[54–56]. Collectively, our results support the possibility that ZIKV causes different pathologies in males and females, although additional NHP studies with larger cohorts of mixed male and female groups are needed to confirm these findings and fully define the contribution of sex to altered immunity and pathogenesis in ZIKV infection. Nevertheless, the possibility of sex differences suggested by our data and reflected in human infection highlights another similarity of the PTM model to human ZIKV infection and the need to evaluate new prophylactic and therapeutic interventions in both sexes.

In summary, using the pigtail macaque model for ZIKV infection, we define the innate immune cells responding to both early and later stages of ZIKV infection in blood, lymph nodes, and mucosal tissues. To date, few studies have explored innate immune cell dynamics of ZIKV infection in NHPs or humans[36], in particular these studies have primarily focused on responses in the blood. By investigating these responses in both blood and tissues, our studies expand upon these findings to demonstrate that early cellular responses in blood correspond to greater viral dissemination to tissues and increased viral persistence in lymph nodes. Innate immune cells responded differently in blood, lymph node, and rectum, despite the identification of ZIKV RNA at all of these sites, and CD16+ monocytes and/or monocyte-derived cells are targets of infection in the blood and lymph node and may mediate viral dissemination into tissues within the host. Therefore, our findings in the PTM will help to inform future studies on ZIKV pathogenesis and disease. Our results show remarkable parallels to outcomes in humans, suggesting that the PTM could provide a valuable pre-clinical model to evaluate the efficacy of ZIKV vaccines and therapeutics in humans. Several

ZIKV vaccine strategies have been advanced to clinical trials[57] based on the success in an NHP model[21,58], and although the value of these models including the PTM reported here in predicting the outcomes in humans is pending the results of these trials, the close resemblance of NHPs to humans in anatomy, immunology, and physiology, as well as our results reported here, that demonstrate close similarities between the PTM and humans in ZIKV pathogenesis, strongly indicate that this model is highly relevant for testing the efficacy of ZIKV vaccines and therapeutics. Indeed, NHP models are considered the gold standard for pre-clinical testing of most vaccines and therapeutics for a broad range of infectious diseases including HIV[59,60], influenza[61], filoviruses, and other flaviviruses[62,63]. Our study also shows significant differences in the immune response to ZIKV and viral distributions and persistence between males and females that highlight the need to investigate ZIKV infection and interventions separately, in both males and females, and suggests possible mechanisms that may drive distinct pathogenesis outcomes of ZIKV infection in different sexes. Finally, the mechanism contributing to differences in ZIKV persistence, dissemination, and sexual transmission is still not clearly understood, but our results also show a possible key role for non-classical monocytes as a primary target of ZIKV infection in vivo in an NHP model. Monocytes are mobile carriers that are capable of crossing tissue barriers and disseminating ZIKV in the host. Importantly, these results suggest that monocyte subsets may have multiple roles during ZIKV pathogenesis in promoting cellular dissemination of viral infection and in producing an inflammatory environment that could lead to viral persistence. The distinct cellular tropism of ZIKV infection for this cell subset in the PTM model may provide an explanation for ZIKV tropism and persistence in certain tissues, and guide future development of interventions for the prevention and treatment of ZIKV infection in humans.

## Methods

**Experimental design and ZIKV challenge**. Eight pigtail macaques (Cohort 1: 4 non-pregnant females, 7–10 years, 6–10 kg; Cohort 2: 4 males, 5–6 years, 10–12 kg) were infected subcutaneously into the right and left forearm over 4–5 sites (females: 4; males: 5) with a total inoculum of $5 \times 10^5$ PFU of a Brazilian Zika isolate (Brazil_2015_MG, GenBank: KX811222.1). The ZIKV inoculum was based on previous NHP studies and the subcutaneous route of administration over several sites was implemented to mimic the natural infection route of a probing mosquito, and this administration had been successful in NHPs[6,10,64,65]. Blood was collected at baseline and periodically throughout the course of infection (every 1–3 days). Whole lymph nodes and mucosal tissues biopsies (endoscopic jejunal and colonic biopsies, as well as pinch biopsies of rectum and vagina) were collected at baseline and weekly after ZIKV infection (Fig. 1a). All animals were pre-screened for the presence of antibodies for West Nile, Dengue, and Zika viruses, and all were negative except a single female animal (L07226), who was seropositive for West Nile virus.

**Care and use of pigtail macaques**. All animals used in this study were housed at the Washington National Primate Research Center (WaNPRC), as accredited by the American Association for Assessment and Accreditation of Laboratory Animal Care (AAALAC) International. The University of Washington's Institutional Animal Care and Use Committee (IACUC) approved all experiments performed on the pigtail macaques, and experiments were in compliance with the U.S. Department of Health and Human Services Guide for the Care and Use of Laboratory Animals and Animal Welfare; however, female animals had weight loss and anemia at 7 dpi that was likely associated with the ZIKV infection[66,67] and resulted in a 1–3 ml blood overdraw relative to IACUC guidelines of 10 ml/kg/ week. Animals were housed in clean and adequate-sized cages, which were sanitized in mechanical cage washers at least every two weeks, with pans cleaned daily. Animals were given environmental enrichment throughout the study, depending on social compatibility. Veterinary staff and animal technicians checked on the animals daily throughout the course of the study to monitor animal care and welfare. For timepoints involving only blood collection or ZIKV inoculation, animals were sedated with ketamine 10 mg/kg intramuscularly (Ketaset® Henry Schein). For timepoints involving PLN and mucosal tissue biopsies, animals were sedated with an intramuscular injection of ketamine 10 mg/kg and dexmedetomidine 0.015 mg/kg (Dexdomitor® Zoetis Inc.). Following sedation, animals were intubated and maintained on isoflurane gas (IsoThesia™ Henry Schein). Animals

received intravenous isotonic fluids and heat support during biopsy procedures, and vital signs and anesthetic depth were monitored continuously (jaw tone, heart rate, respiration rate, SpO$_2$, ETCO2, non-invasive blood pressure, and body temperature). Jejunal and colonic mucosal tissues were collected via endoscopic biopsy, and vaginal and rectal tissues were collected under direct visualization with biopsy forceps. Peripheral LNs were collected surgically from axillary or inguinal sites using aseptic technique. Animals received buprenorphine sustained release 0.2 mg/kg subcutaneously (Buprenorphine SR™ SR Veterinary Technologies) immediately prior to biopsy procedures, and bupivacaine local blocks were provided at each lymph node biopsy site (Marcaine™ Hospira Inc.). Animals were given atipamezole 0.15 mg/kg intramuscular (Antisedan® Zoetis Inc.) following biopsy procedures to reverse dexmeditomidine sedation. Macaques used for infectious disease studies are nearly universally sedated with ketamine for procedures to ensure staff safety and permit collection with minimal stress to the animals. It is therefore also the best choice to permit comparison to the other literature generated on ZIKV in macaques, which also used repeated ketamine sedation during their studies[6–8]. Prior to necropsy, animals were sedated with ketamine and dexmeditomidine as described above and euthanized with an overdose of intravenous pentobarbital sodium (Euthasol® Virbac Corp.) in accordance with the 2013 Edition of the American Veterinary Medical Association (AVMA) Guidelines for the Euthanasia of Animals. All animals reached the experimental endpoint without becoming severely ill.

**Sample collection and processing.** Peripheral blood mononuclear cells were isolated from pigtail macaque blood by Histopaque-1077 (Sigma-Aldrich) density gradient in Accupsin conical tubes (Sigma-Aldrich). Red blood cells were lysed using ACK lysis buffer (ThermoFisher) and then resuspended in RPMI-1640 media (Invitrogen) supplemented with 10% Fetal Bovine Serum (FBS) (VWR), 50 µg/ml Gentamicin sulfate (ThermoFisher), and 5 mg/ml of Penicillin–Streptomycin–Glutamine (Pen–Strep–Glut) (ThermoFisher). Intraepithelial and lamina propria lymphocytes of gut and vaginal mucosa biopsies were isolated by straining through a 70-micron filter (Corning) following a 1 hour (h) enzymatic digestion at 37 °C with 40 µg/ml Liberase TM (Sigma-Aldrich) and 80 µg/ml DNase (Sigma-Aldrich) in RPMI-1640 media supplemented with 5 mg/ml of Pen–Strep–Glut (ThermoFisher). Lymphocytes from whole lymph node biopsies were isolated by directly straining through a 70-micron filter in RPMI-1640 media supplemented with 10% FBS (VWR) and 5 ml of 1X Pen–Strep–Glut (ThermoFisher), and 50 µg/ml Gentamicin sulfate (ThermoFisher).

**Cell lines and virus.** Vero cells were obtained from the World Health Organization and C6/36 *Aedes albopictus* cells were obtained from ATCC cultured in Dulbecco's Modification of Eagle's Medium (DMEM, Cellgro) supplemented with 10% FBS (HyClone), 2 mM L-Glutamine, 1 mM Sodium pyruvate, 100 U/ml of Penicillin, 100 µg/ml of Streptomycin, and 1X MEM Non-essential Amino acid solution (Sigma-Aldrich). The World Reference Center of Emerging Viruses and Arboviruses (Galveston, Texas, USA) provided the ZIKV strain isolated in Brazil (Brazil_2015_MG, GenBank: KX811222.1 https://www.ncbi.nlm.nih.gov/nuccore/KX811222.1). Working stocks were obtained by plaque-purifying virus and amplified once in C6/36 cells. Virus was adsorbed to cells in DMEM supplemented with 1% FBS at 37 °C. After the 2-h incubation, the inoculum was removed, and virus was propagated in complete media for 6 days, with media changed at 3 dpi. Supernatants at 6 dpi were then collected and centrifuged at 2000 RPM at 4 °C for 10 min and frozen in aliquots at −80 °C. Vero cells were inoculated with tenf-old serial dilutions of viral stock and incubated for 2 h at 37 °C, after which a 1% agarose overlay was added. Five days later, a second 1% agarose overlay containing 2% Neutral Red (Sigma-Aldrich) was added to the cells. Plaques were counted 4 h after the addition of the Neutral Red overlay.

**Measurement of ZIKV RNA load.** Viral RNA load was assessed in the plasma and tissue biopsies (female: 1 biopsy; male: 2 biopsies) using a ZIKV-specific RT-qPCR assay, as previously described[10]. Whole blood was collected into EDTA tubes and spun at 931 × g at room temperature for 15 min. Plasma was isolated from the top layer and aliquots stored at −80 °C until further processing. Tissues were placed into RNAlater Stabilization Solution (ThermoFisher Scientific) for 24 h at 4 °C and then transferred to −80 °C until further processing. For the analysis of ZIKV RNA in tissue, organs were weighed and homogenized in TRIzol Reagent (Life Technologies) using a bead-beater apparatus (Precellys, Berlin Corp.). RNA was extracted from the tissue homogenates using a RNeasy Mini Kit (Qiagen). For analysis of ZIKV RNA in plasma, RNA was isolated using the QIAamp Viral RNA Mini Kit (Qiagen). Following RNA extraction, cDNA was synthesized from 400 ng of total RNA extracted from tissue and 10 µl of total RNA extracted from plasma using the iScript Select cDNA Synthesis Kit (Bio-Rad) according to the manufacturer's protocol for gene-specific primers. Viral RNA was quantified using the TaqMan Universal PCR Master Mix (Applied Biosystems) and a 7300 Real-Time PCR System (Applied Biosystems), using primers defined in Supplementary Table 5 and as previously described[68] that are conserved in the Brazil Fortaleza genome. The ZIKV-specific primer/probe set was synthesized by Integrated DNA Technologies, with 5-FAM as the reporter dye for the probe. To adhere to stringent

guidelines, cycle threshold (Ct) values >38 were deemed as not reliably detected and were not reported. Ct values <38 in at least two of the triplicates and falling within the standard curve determined from diluted known quantities of ZIKV genome are reported. Copy number sensitivity ranged from 10 to 83 copies/qPCR reaction across plates.

**Immunophenotyping.** Isolated PBMCs, gut lymphocytes, and lymph node cells were assessed for viability with a live/dead stain (Life Technologies) and stained with a panel of antibodies, details described in Supplementary Table 6, in brilliant stain buffer (BD Biosciences) to identify innate immune cells. To identify DCs in whole blood, cells were surface stained with non-DC antibodies, blood cells were then FACS lysed (BD Biosciences) and surface stained with DC antibodies. Cells were then resuspended in 1% paraformaldehyde and samples were acquired on a LSRII (BD Biosciences) using FACS Diva software (version 8). Samples were analyzed using FlowJo software version 9.9.4 (FlowJo, LLC) and gating is depicted in Supplementary Fig. 8. All events were first gated on FSC singlets, CD45$^+$ leukocytes, live, and then cells according to FSC-A and SSC-A profiles. Dendritic cells were gated as CD20$^−$CD3$^−$HLA-DR$^+$CD14$^−$ and then divided into pDCs (CD123$^+$CD11c$^−$) and mDCs (CD123$^−$CD11c$^+$). Monocytes were identified as CD20$^−$CD3$^−$HLA-DR$^+$CD14$^±$CD16$^±$, and then categorized into classical (CD14$^+$CD16$^−$), intermediate (CD14$^+$CD16$^+$), and non-classical subsets (CD14$^{int}$CD16$^+$). Neutrophils were identified as CD3$^−$CD11b$^+$CD14$^+$ and then selected for high SSC-A[69]. NK cells were identified as CD14$^−$CD3$^−$NKG2A$^+$CD8α$^+$. Innate cell gates that were unable to meet minimum threshold of ≥100 cells/gate were excluded. Cellular activation of monocytes and DC subsets was measured using CD86 mean fluorescence intensity (MFI) that met a minimum threshold of ≥100 cells/gate. To identify intracellular NS3, cells were fixed, permeabilized (BD Biosciences) and stained with anti-WNV NS3 (R&D Systems; BAF2907), which was confirmed to cross-react with ZIKV-NS3 using an in vitro infection system and then with FITC Streptavidin (BioLegend). NS3-positive cells were identified after background and pre-infection subtraction and meeting a cellular threshold (≥100 cells/gate).

**Assessment of plasma cytokines and chemokines.** Cytokine and chemokine levels in plasma were analyzed using a Milliplex MAP (multi-analyte profiling) nonhuman primate cytokine magnetic bead panel premixed 23-plex kit (Millipore). The levels of the analytes were assessed on a Bio-Plex 200 system (Bio-Rad).

**Statistical analysis.** Non-parametric statistical methods were employed for all comparisons. Specifically, Wilcoxon tests were used to compare continuous values across sex groups, paired Wilcoxon tests were used to evaluate cell fraction differences to baseline at each timepoint (Fig. 2 and Supplementary Fig. 2–4 and 6), and Spearman's rank-transformed correlation analyses were conducted to evaluate correlation robustly. All analyses were conducted using two-sided tests at the 0.05 level. No adjustment was performed to control for the multiple hypotheses tested. Values based on below-threshold cell counts are considered missing values for the analysis. A minimum of $n = 5$ non-missing values per group was required to conduct two-sample tests, and a minimum of $n = 5$ non-missing paired difference values was required to conduct one-sample tests. Analyses were conducted in Prism version 6.0 h (GraphPad) and in R version 3.3.2[70].

**Data availability.** The data that support the findings of this study are available from the corresponding author upon reasonable request.

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

## Acknowledgements

We thank all members of the Fuller laboratory for their technical support and helpful discussion. We thank B. Agricola, S. Wangari, D. May, J. Ahrens, N. Iwayama, and J. Lane for their expert support in animal husbandry and care. We also thank T. Hensley-McBain and J. Manuzak for technical support in developing the flow cytometry panels. This work was supported in part by funds from the National Institute of Health (P51-OD010425) and in part by start-up funds from WaNPRC and UW to N.R.K. M.A.O. is supported by T32-AI007140.

## Author contributions

M.A.O., J.T.G., C.J.M., D.B., N.R.K., M.G., and D.H.F. designed and coordinated the studies. M.A.O. and T.B.L. led the immunological studies. J.T.G. led the virologic assays. C.J.M. conducted the cytokine and chemokine assays. M.A.O. led the analysis of immunological, cytokine and chemokine analysis. M.A.O. and J.T.G. analyzed the data. P.T.E led the statistical analysis. C.M. and J.S. led the clinical care of the animals. M.A.O. and D.H.F. led the studies, interpreted the results, and wrote the paper with all co-authors.

## Additional information

**Competing interests:** The authors declare no competing interests.

