## [Peer Review File · Nature Communications]

Editorial Notes:

Parts of this peer review file have been redacted as indicated to maintain the confidentiality of unpublished data.

This manuscript has been previously reviewed at another journal that is not operating a transparent peer review scheme. This document only contains reviewer comments and rebuttal letters for versions considered at Nature Communications.

Reviewers' comments:

Reviewer #1 (Remarks to the Author):

The re-submission by O'Connor et al has addressed many of the concerns previously stated. The authors show virological and immunological data not only from the blood, but also from multiple mucosal tissues, with analyses comparing these responses between males and females. However, many of the key findings (from the blood) have been reported previously. We suggest that the submission be modified to focus on only the novel aspects, which are the findings from the tissues and the comparisons between males and females.

Major suggestions

1) A number of aspects of this submission have been reported elsewhere:

a. Changes in plasma cytokines shown in Fig. 3a, including MCP-1, have already been reported by others (Hirsch et al. *PloS Pathog* 2017; Osuna et al., *Nat. Med.*, 2016), as stated by the authors in line 199.

b. Michlmayr et al., *Nat. Microbiol* 2017, has already demonstrated in acute ZIKV infection in humans that CD14⁺CD16⁺ monocytes, followed by mDCs, harbor the vast majority of ZIKV among blood cells in vivo. This was also acknowledged by the authors in line 278.

2) The stated focus of this manuscript is on the early innate immune response to ZIKV. However, the earliest tissue timepoint sampled is day 7 post-infection, after the resolution of plasma viremia.

3) The authors should clarify the relevance of viral persistence in the tissues examined (rectum, jejunum, and colon). Is the virus still infectious at this stage and thus transmissible? Infiltration to, and persistence in the male genital tract and CNS were not examined. For a study of ZIKV persistence, these are probably the two most important tissues to sample.

4) There is no consideration of monocyte differentiation into macrophages in the tissues. Is the definition of monocytes as CD3⁻CD20⁻HLADR⁺CD16⁺/-CD14⁺/- sufficient for identification of monocytes in mucosal tissues and to distinguish them from mature macrophages or dendritic cells? Did the authors conduct any evaluation of macrophages in the tissues?

5) In the original submission, one male monkey was positive in the peripheral LN at the pre-infection timepoint. This is no longer the case. Was the sample re-evaluated?

6) Caution should be used in interpreting AUC calculations between males and females. There are some parameters for which males and females have different values at baseline and these differences may, at least partially, drive the differences in AUC between males and females. For example, the higher blood neutrophil AUC frequencies observed in females in Fig. 2d are likely affected by the fact that all of the females have higher neutrophil frequencies before infection than males.

7) Line 175: What data supports the statement that females have delayed neutrophil recruitment to the blood? In Fig. 2d, the increase in blood neutrophils appears to occur on the same day (day 9) for both males and females, although it is more dramatic in the females.

Minor suggestions

- Baseline ZIKV IgG/IgA antibodies should be shown in a supplemental figure.
- Fig. 1b: It appears that values of some of the LOD lines on the graphs have changed since the original submission but stated LOD remains the same.
- Fig. 1b: This reviewer suggests using the same y-axis scale for all graphs.
- Line 111: change the word "cute" to "acute"
- Figure 2: The "a" indicating part a is missing. It seems that text indicating which graphs are whole blood, PLN, and rectum are also missing. The AUC graph in part a is missing a y-axis title.
- Fig. 2 legend: neutrophils are stated as being defined as CD3-CD11b+CD14+HLA-DR-, although the Methods section defines them as CD3-CD11b+CD14+HLA-DR+. Please correct.
- Can the authors provide a citation to justify their definition of neutrophils as CD11b+CD14+SSC-A-high? Using this cell surface marker combination to define neutrophils is not conventional.
- Line 191: What is meant by "corresponded with higher levels of pDCs in the blood"? Was a correlation analysis performed? What were the correlation coefficients (negative or positive association)? These questions also apply to the sentence starting on line 192 regarding neutrophils.
- Line 270: The statement implies that blood pDC levels correlate with monocytes. However, this correlation was not shown. This is also true for the following statement that monocytes correlate with viral persistence in the LN.
- Supplemental Figure 2: Only highlight statistically significant values.
- Supplemental Figure 1a and Supplementary table 3: Why is there insufficient data points for stats on CD86 MFI on pDCs when data points are available for all other immune subsets? Are data being excluded? Also, the median line at day -10 appears to be higher than all data points shown. Unless data points are excluded from being shown, this cannot possibly be the median.

Reviewer #2 (Remarks to the Author):

General:

Well written paper. Interesting results and novel to the field. A good first look at how the innate immune system responds to infection.

Abstract:

"eight male and female adult pigtail macaques"

- It would clarify to mention 7 flavi-naive PTMs and 1 primed to WNV.

Introduction:

"Clinical diagnosis of ZIKV infection"

- Early in the course of infection may be a reasonable qualifier for this statement.
- A statement discussing the relevance of the findings to the bigger picture of pathogenesis or

countermeasure development would strengthen the paper.

Results:

"Eight pigtail macaques"

- WNV positivity is mentioned in one animal, it would add clarity to have a single sentence in the results section explaining what screening was performed and the results.

Discussion:

Methods:

"Cohort 1: 4 non-pregnant females, 7-10 years, 6-10 kg; Cohort 2: 4 males, 5-6 years, 10-12 kg

- Could the difference in age or weight account for any of the differences noted which appeared to be along gender lines?

"over 4-5 sites"

- Was this in an attempt to mimic a probing mosquito? A statement explaining the rationale would strengthen the manuscript.

"mimic the natural infection route of a mosquito bite"

- I would think an ID delivery would be more consistent with the anatomic space probed by a proboscis, do the authors have a different perspective?

"presence of antibodies to West Nile, Dengue, and Zika"

- Clear. I assume no risk of other flavivirus exposures either from natural circulation or past direct or indirect exposure in the context of research?

"ketamine 10 mg/kg intramuscular"

- Is there literature on impact of anesthesia on immune responses such as cytokine and chemokine secretion?

What are the major claims of the paper?

- Innate immune signals following Zika virus infections measured in male and female NHPs vary across PTM gender and time of infection.

Are they novel and will they be of interest to others in the community and the wider field?

- I believe yes to both.

Is the work convincing, and if not, what further evidence would be required to strengthen the conclusions?

- The work is convincing but requires randomization of PTMs to ensure equal age and weight and gender assigned to the various groups.

Reviewer #1

The re-submission by O'Connor et al has addressed many of the concerns previously stated. The authors show virological and immunological data not only from the blood, but also from multiple mucosal tissues, with analyses comparing these responses between males and females. However, many of the key findings (from the blood) have been reported previously. We suggest that the submission be modified to focus on only the novel aspects, which are the findings from the tissues and the comparisons between males and females.

*Response: To gain a better understanding of Zika virus disease progression, we believe it's imperative to investigate responses in both the blood and tissues, not only to better understand the dynamics of virological-host interactions in these tissues separately, but also to determine the relationship between responses in these compartments. For this reason, we feel that the novelty of our manuscript is not limited to responses in the tissue but additionally, in the comparison between responses in the blood vs. tissues. Specifically, our concurrent analysis of responses in both compartments was key to revealing new insights into ZIKV disease progression. For example, we demonstrate in **Figure 5** that early cellular innate immune responses in the blood corresponded with greater viral persistence in the lymph node and increased viral dissemination to the rectum. Furthermore, in **Figure 4**, we show that the CD16⁺ non-classical monocytes are the major targets of ZIKV infection in the blood as well as the peripheral lymph node, demonstrating that these cells may drive dissemination of virus into the tissues. In addition to providing new insight on ZIKV pathogenesis, our data demonstrate correlations between responses in the blood and tissues. To further emphasize this, we included the following in the discussion (**Lines 377-385**): "To date few studies have explored innate immune cell dynamics of ZIKV infection in NHPs or humans, in particular these studies have primarily focused on responses in the blood. By investigating these responses in both blood and tissues, our studies expand upon these findings to demonstrate that early cellular responses in blood correspond to greater viral dissemination to tissues and increased viral persistence in lymph nodes. Innate immune cells responded differently in blood, lymph node and rectum despite identification of ZIKV RNA at all of these sites and CD16⁺ monocytes and/or monocyte-derived cells are targets of infection in blood and lymph node and may mediate viral dissemination into tissues within the host."*

Major Concerns

1. A number of aspects of this submission have been reported elsewhere:
 - a. Changes in plasma cytokines shown in Fig. 3a, including MCP-1, have already been reported by others (Hirsch et al. PloS Pathog 2017; Osuna et al., Nat. Med., 2016), as stated by the authors in line 199.
 - b. Michlmayr et al., Nat. Microbiol 2017, has already demonstrated in acute ZIKV infection in humans that CD14⁺CD16⁺ monocytes, followed by mDCs, harbor the vast majority of ZIKV among blood cells in vivo. This was also acknowledged by the authors in line 278.

Response: In the development of non-human primate models to study human ZIKV infection it is important to validate findings from other NHP species models and to identify similarities between ZIKV infection in humans and NHPs. It is also important in the development of different ZIKV animal models to identify whether the strain of ZIKV infection, route, and dose of administration modulate different aspects of the disease. In particular, it was previously noted that different routes of infection (subcutaneous vs.

intravenous) and different ZIKV strains (French Polynesia vs. Brazil) may influence the kinetics and magnitude of ZIKV viral replication in rhesus macaques and may consequently impact the development of immune responses to the infection (Coffey et al., PLOS One, 2017).

As this manuscript is the first to describe ZIKV infection in the PTM, our immunological findings are novel in the context of this model and reveal new insights and parallels to human infection not previously reported or observed in other NHP species. For example, in comparison to the RM, we observed similar changes in cytokine profiles, including increases in IL-1RA (Hirsch AJ et al. PloS Pathogens, 2017) and MCP-1 (Hirsch AJ et al. PloS Pathogens, 2017 ; Osuna CE et al. Nat Med, 2016). However, we also describe trends that were unique in the PTM including plasma sCD40L and IL-8 increases in the PTM versus decreases in the RM (Osuna CE et al. Nat Med, 2016). These comparisons are described in the discussion (Lines 321-326): “The host response and viral dynamics following ZIKV infection in PTM reported here, including a marked increase in monocyte frequency in the blood and increased production of pro-inflammatory cytokines/chemokines (IL-1RA, MCP-1, and IL-15) during peak viremia closely resembles responses to ZIKV infection previously reported in RMs, but differs in other measures including enhanced plasma sCD40L and IL-8 and lack of sporadic plasma viral blips following viral clearance at 7 dpi.” Future evaluation of Zika virus infection in humans is needed to determine which aspects of the NHP disease resembles humans in these responses.

We recognize that monocytes and mDCs were identified as ZIKV viral cellular targets in humans (Michlmayr D et al., Nat Microbiol, 2017; Foo SS et al., Nat Microbiol, 2017), but our report is the first to confirm that these specific cell types are also the targets during ZIKV infection in NHPs and to identify the non-classical monocytes as a primary cellular target in blood and in tissues. These findings provide new insight into the parallels between humans and NHPs in ZIKV infection. Additionally, we demonstrate, that early cellular innate immune responses, including pDC recruitment to the blood and production of MCP-1, correlate with ZIKV persistence in the lymph node or mucosal tissues demonstrating, for the first time, a role for specific innate responses in ZIKV pathogenesis and persistence. Specifically, the production of MCP-1 and pDC recruitment may create an inflammatory environment, lead to further recruitment of inflammatory cells and targets of ZIKV infection. As such, our findings are unique in not only demonstrating the PTM as a valuable model of human ZIKV infection but providing new insight into the mechanisms of ZIKV pathogenesis. To emphasize these points, we added the following statement(s) to the discussion (Lines 284-292): “These results are consistent with other studies in the rhesus macaque that showed ZIKV infection in innate immune cells of myeloid and neutrophil origin in the spleen and lymph nodes, and studies in humans that showed ZIKV infection of monocytes in the blood, but extend these findings to identify, for the first time, that CD16⁺ monocytes are preferential early targets of ZIKV infection in both the blood and peripheral lymph nodes. These findings provide new insight into the mechanisms of ZIKV pathogenesis and establish the PTM as a highly relevant model for evaluating ZIKV infection in humans. These results are also the first to show in NHPs preferential infection of non-classical monocytes and to identify these cells as viral targets in both the blood and peripheral lymph node.”

2. The stated focus of this manuscript is on the early innate immune response to ZIKV. However, the earliest tissue timepoint sampled is day 7 post-infection, after the resolution of plasma viremia.

Response: The manuscript reports early innate immune responses in the blood (days 1-4) and strikingly, shows that certain responses in the blood significantly correlated with viral tissue tropism and persistence in these tissues at day 7 when viral replication had resolved in the blood. This indicates a role for these early innate responses in ZIKV pathogenesis. We avoided tissue sampling during peak viral replication prior to day 7 to avoid the possibility of tissue collection influencing pathogenesis in the tissues and as such, cannot draw conclusions about the innate immune dynamics in the tissues. However, the correlations between the persistence of ZIKV RNA in these tissues (lymph node, rectum) at 7 dpi and cellular innate responses occurring in the blood at days 1-4, provide new insight into a possible mechanism of ZIKV persistence and/or dissemination in the tissues as detailed above in our response to Point 1 above. We have clarified this point throughout the manuscript to emphasize that the early (pre-day 7) innate immune events were limited to the blood. It is important to note, however, that although some of our analysis of innate responses in the blood had returned to baseline by day 7, in the tissues, significant changes in the frequency of innate cells in the lymph nodes and mucosal tissues occurred 7-21 dpi. In the future we would like to evaluate cellular innate immune responses in the tissues prior to day 7 but are unable to evaluate this in our current studies.

3. The authors should clarify the relevance of viral persistence in the tissues examined (rectum, jejunum, and colon). Is the virus still infectious at this stage and thus transmissible? Infiltration to, and persistence in the male genital tract and CNS were not examined. For a study of ZIKV persistence, these are probably the two most important tissues to sample.

[redacted]

4. There is no consideration of monocyte differentiation into macrophages in the tissues. Is the definition of monocytes as CD3-CD20-HLADR+CD16+/-CD14+/- sufficient for identification of monocytes in mucosal tissues and to distinguish them from mature macrophages or dendritic cells? Did the authors conduct any evaluation of macrophages in the tissues?

Response: For evaluation of monocytes in tissues, we utilized a consistent gating scheme for monocytes identified in the blood. We recognize that migrating monocytes can differentiate into macrophages or dendritic cells upon entrance into tissues. Because of

this reason, we specifically excluded dendritic cells (CD14⁺) prior to our analysis of monocytes in the tissues. Unfortunately, our panel lacked markers needed for resolution of tissue macrophages e.g. CD163 (Wonderlich ER, et al. J of Immunology, 2015) or CD68 (Ortiz AM, et al. J of Virology, 2015). We have added the following text to the results, to address these concerns (Lines 169-171): “In the lymph node and mucosal tissues, we identified monocytes using the same gating scheme as in the blood, which excluded dendritic cells, but may include monocyte-derived and macrophage cell populations.”

5. In the original submission, one male monkey was positive in the peripheral LN at the pre-infection timepoint. This is no longer the case. Was the sample re-evaluated?

Response: Based on previous Reviewer concerns, we re-evaluated the Z11157 PLN -10 dpi sample prior to transfer of this manuscript to Nature Communication.

*We repeated the ZIKV qRT-PCR assay for all male and female PLN samples with a primer and probe set that detects all known genotypes of Zika virus (ZIKV 1087/1163c/1108FAM). Based on the standard curve, the assay limit of detection of the repeated experiment was 25 copies, with an average cycle threshold (Ct) cut-off of 36.7. Samples reported positive had Ct values ≤ 36.7 for at least two of the three replicates in the assay. The assay results showed baseline PLN (-10 days post-ZIKV inoculation) samples were negative for ZIKV RNA in all animals. In particular, the average Ct value of Z11157 PLN -10 dpi was 38.4 indicating it was above the average Ct threshold cut-off and therefore negative before infection. Therefore, in our resubmitted manuscript, we updated the data for Z11157 in **Figure 1b** to reflect these data and also updated the area under the curve analysis (AUC) of the viral burden within the PLN (**Figure 1d**, **Figure 5c**). The results in our repeated assay show that all animals were negative for ZIKV detection at baseline, a result that is consistent with lack of detectable ZIKV-specific antibody responses prior to infection (see new **Supplementary Figure 1**).*

6. Caution should be used in interpreting AUC calculations between males and females. There are some parameters for which males and females have different values at baseline and these differences may, at least partially, drive the differences in AUC between males and females. For example, the higher blood neutrophil AUC frequencies observed in females in Fig. 2d are likely affected by the fact that all of the females have higher neutrophil frequencies before infection than males.

Response: We appreciate this critique and understand the limitation of this sort of analysis. We chose to evaluate AUC of immune responses for several reasons 1) innate immune cells peaked at different timepoints, 2) not all animals peaked at the same timepoint, and 3) some parameters had a sustained peak response. Therefore, we felt that using AUC was the most faithful way to consistently present the data without biasing towards a specific timepoint. We agree with the Reviewer that in the context of blood neutrophils, higher frequencies of neutrophils in the females prior to infection likely affected the AUC analysis. Additionally, differences in baseline frequencies of certain innate immune cells at different sites could influence the kinetic and magnitude of the host response to ZIKV infection, for this reason we have added a point about this in the

discussion which also addresses comments from Reviewer #2 below (Methods 1 and 5) (Lines 355-358): “In addition, other factors such age, weight, repeated ketamine sedation, and inherent differences in cellular frequencies prior to infection could alternatively influence cellular immune responses and associated sex differences.”

7. Line 175: What data supports the statement that females have delayed neutrophil recruitment to the blood? In Fig. 2d, the increase in blood neutrophils appears to occur on the same day (day 9) for both males and females, although it is more dramatic in the females.

Response: We agree with the conclusions made by the Reviewer and agree that our wording was unclear. We have changed the text to “Together these results show, especially in the females, early neutrophil recruitment to mucosal sites during acute infection and delayed neutrophil recruitment to blood in all animals following viral clearance.” (Lines 183-185).

Minor Suggestions

1. Baseline ZIKV IgG/IgA antibodies should be shown in a supplemental figure.

*Response: New **Supplementary Figure 1** (see below) has been added to the manuscript to address this, with the associated text in the Results section (Lines 113-118): “L07266 cleared virus in the peripheral lymph node (PLN) by day 7, more quickly than the other animals. However, baseline anti-ZIKV envelope IgG or IgA antibodies were not higher in this animal (**Supplementary Figure 1**) and baseline and acute levels did not correspond to ZIKV viral burden in the PLN of L07266 (data not shown) indicating pre-existing immunity to WNV did not contribute to the early viral clearance in the PLN in this animal (Fig. 1b, c).”*

New Supplementary Figure 1

2. Fig. 1b: It appears that values of some of the LOD lines on the graphs have changed since the original submission but stated LOD remains the same.

Response: The level of detection (LOD) lines were described accurately in the figure legends, and we thank the Reviewer for pointing out the inconsistencies in the figure. We have changed the figures accordingly.

3. Fig. 1b: This reviewer suggests using the same y-axis scale for all graphs.

*Response: We have heeded the advice of the Reviewer and have put all graphs of **Figure 1b** onto the same y-axis scale.*

4. Line 111: change the word “cute” to “acute”

Response: This has been changed in the text.

5. Figure 2: The “a” indicating part a is missing. It seems that text indicating which graphs are whole blood, PLN, and rectum are also missing. The AUC graph in part a is missing a y-axis title.

*Response: We thank the Reviewer for identifying this, these missing parts have been added back to **Figure 2**.*

6. Fig. 2 legend: neutrophils are stated as being defined as CD3⁻CD11b⁺CD14⁺HLA-DR⁻, although the Methods section defines them as CD3⁻CD11b⁺CD14⁺HLA-DR⁻. Please correct.

Response: We thank the Reviewer for identifying this inconsistency and have edited the definition of neutrophils to CD3⁻CD11b⁺CD14⁺HLA-DR⁻ throughout the manuscript.

7. Can the authors provide a citation to justify their definition of neutrophils as CD11b⁺CD14⁺SSC-A^{high}? Using this cell surface marker combination to define neutrophils is not conventional.

*Response: We have now included a citation (Hensley-McBain et al. 2018) (#74) to the methods (**Line 531**) to justify identification of neutrophils as CD3⁻CD11b⁺CD14⁺HLA-DR⁻SSC-A^{Hi} in non-human primates.*

8. Line 191: What is meant by “corresponded with higher levels of pDCs in the blood”? Was a correlation analysis performed? What were the correlation coefficients (negative or positive association)? These questions also apply to the sentence starting on line 192 regarding neutrophils.

Line 270: The statement implies that blood pDC levels correlate with monocytes. However, this correlation was not shown. This is also true for the following statement that monocytes correlate with viral persistence in the LN.

Response: We apologize for these confusions and have provided greater detail of the methods used and/or the reference figures that were used in order to make these conclusions:

*Original Line 191, new **Line 199-201**: “In particular, males had higher MCP-1 levels at 2 dpi (**Fig. 3b**, $p=0.0286$), and plasma MCP-1 levels positively correlated with higher levels of pDCs in the blood at this timepoint ($p=0.0458$, $r=0.7381$, data not shown).”*

*Original Line 192, new **Line 201-204**: “In contrast, the neutrophil chemoattractant interleukin 8 (IL-8) was significantly higher in female plasma a 2 dpi (**Fig. 3b**, $p=0.0286$)”*

and higher plasma IL-8 positively correlated with increased neutrophil recruitment to the blood at 9 dpi ($p=0.0154$, $r=0.833$, data not shown).”

Original Line 270, new **Line 278-281**: “Here, we further extend these findings and show that increases in blood pDCs and plasma MCP-1 in the first few days of infection are associated with greater viral burden in the rectum and PLN and promote recruitment of non-classical monocytes to the lymph node (**Fig. 5**).”

Original Line 271, new **Line 281-283**: “Notably, we show that monocytes serve as early and primary cellular targets of in vivo ZIKV infection in the blood and lymph node and correlate with great viral load at these sites (**Fig. 4**).”

9. Supplemental Figure 2: Only highlight statistically significant values.

Response: For all supplementary tables (1-3), we agree that our display of significant ($p \leq 0.05$) versus trending values ($p \leq 0.08$) may have been misleading. Given that our sample size is low, we felt that 0.05 was too strict of a threshold for including information about the directionality of the effect. Direction of even trending effects is useful information for a reader, but we intend to be very clear about which are significant and which are not. In response to the Reviewer, we have now bolded and italicized all significant values and have italicized trending p-values. The colored text indicates directionality, rather than significance, therefore we have clarified both of these points in the Supplemental figures legends: Example Supplementary Table 2 legend “with $p \leq 0.05$ indicating significantly different than baseline differences (italicized and bolded) and values indicate trending ($p \leq 0.08$, italicized). Coloring indicates higher (green) or lower (red) after baseline.”

10. Supplemental Figure 1a and Supplementary table 3: Why is there insufficient data points for stats on CD86 MFI on pDCs when data points are available for all other immune subsets? Are data being excluded? Also, the median line at day -10 appears to be higher than all data points shown. Unless data points are excluded from being shown, this cannot possibly be the median.

*Response: For our analysis of innate immune cells we had a minimum cellular threshold of ≥ 100 cells/gate and have clarified this point in the methods as follows (**Lines 531-534**): “Innate cell gates that were unable to meet minimum threshold of ≥ 100 cells/gate were excluded. Cellular activation of monocytes and DC subsets were measured using CD86 mean fluorescence intensity (MFI) that met a minimum threshold of ≥ 100 cells/gate” and (**Lines 560-563**) “Values based on below-threshold cell counts are considered missing values for the analysis. A minimum of $n = 5$ non-missing values per group was required to conduct two-sample tests, and a minimum of $n = 5$ non-missing paired difference values was required to conduct one-sample tests.” Because of the low frequencies of certain cell subsets in certain animals, there was not always a sufficient number of animals to perform statistical analysis. To further clarify this, we have edited the legends of new Supplementary figures 2, 3, and 4 and Supplementary Table legends 2 and 3 to indicate when sufficient values were not available, an example for Supplementary Figure 3 is given: “Statistical analysis in the vagina of certain subsets was not possible (indicated as N/A in **Supplementary Table 2**), because the frequencies of cells in some animals at some timepoints were below the minimum threshold.” We have also fixed the median line in new **Supplementary Figure 1a**.*

Reviewer #2

Abstract

1. "eight male and female adult pigtail macaques" It would clarify to mention 7 flavi-naive PTMs and 1 primed to WNV.

Response: Please see response under "Results", Point 1 below.

Introduction

1. "Clinical diagnosis of ZIKV infection" Early in the course of infection may be a reasonable qualifier for this statement. A statement discussing the relevance of the findings to the bigger picture of pathogenesis or countermeasure development would strengthen the paper.

Response: We have modified the introduction to address these changes (Line 54) and have added the following statement to demonstrate the bigger picture (Lines 94-95): "Additionally, the findings in these studies provide a greater understanding of the early cellular events that may influence ZIKV infection in humans."

Results

1. "Eight pigtail macaques" WNV positivity is mentioned in one animal, it would add clarity to have a single sentence in the results section explaining what screening was performed and the results.

Response: We agree and have added the following statement to the results section (Lines 101-103): "All animals were pre-screened for the presence of antibodies to WNV, DENV, and ZIKV, and all were negative except a single female animal (L07226), who was seropositive for West Nile."

Methods

1. "Cohort 1: 4 non-pregnant females, 7-10 years, 6-10 kg; Cohort 2: 4 males, 5-6 years, 10-12 kg" Could the difference in age or weight account for any of the differences noted which appeared to be along gender lines?

Response: Pigtail macaques are sexually dimorphic (http://pin.primate.wisc.edu/factsheets/entry/pigtail_macaque) and adults males weigh 6.2-14.5 kg and adult females weigh 4.7-10.9 kg, therefore we selected healthy adults for our study. Pigtail macaques sexually mature by age 3 (female) and between 3-4.5 years (male), and macaques are considered aged >18 years old (Coe et al. Age (Dordr), 2012; Haberthur K et al., Experimental Gerontology, 2010), therefore we excluded juveniles and aged PTMs in our studies and selected sexually-mature adults for these studies. The age of the animals used in our studies were also consistent with the age of RMs used in previous ZIKV infection studies: females age 5-13 years and males age 3-11 years (Coffey et al, PLOS One, 2017; Dudley et al, Nat Communications, 2016; Hirsch et al, PLOS Pathogens, 2017). It is possible that some of the inherent gender differences may influence ZIKV pathogenesis and we have added text in the discussion highlighting the need to perform additional studies with larger cohorts of mixed male and female groups (Lines 368-371). In addition, we added the following statement to the Discussion to

address this point, and the point raised below in Methods, Point 5 as well as Major Concern, Point 6 from Reviewer #1 above. (Lines 355-358): “In addition, other factors such as age, weight, repeated ketamine sedation, and inherent differences in cellular frequencies prior to infection could alternatively influence cellular immune responses and associated sex differences.”

2. “over 4-5 sites” Was this in an attempt to mimic a probing mosquito? A statement explaining the rationale would strengthen the manuscript.

Response: We have edited the following explanation in the methods to address this concern (Lines 417-419): “The ZIKV inoculum was based on previous NHP studies and the subcutaneous route of administration over several sites was implemented to mimic the natural infection route of a probing mosquito and this administration had been successful in NHPs”

3. “mimic the natural infection route of a mosquito bite” I would think an ID delivery would be more consistent with the anatomic space probed by a proboscis, do the authors have a different perspective?

Response: ZIKV is primarily transmitted via mosquito bite but can also be transmitted sexually and through the blood. We chose the subcutaneous route, rather than intravaginal, intrarectal, or intravenous infection, to mimic the route of mosquito bite infection. Subcutaneous ZIKV infection had been successfully used NHPs (Adams Waldorf et al., Nature Medicine, 2016; Dudley et al., Nature Communications, 2016; Koide et al., Frontiers in Microbiology, 2016) to mimic mosquito-transmitted ZIKV infection and we wanted our studies to be consistent with published routes of administration. We agree with the Reviewer that intradermal delivery (ID) could be an alternate route of ZIKV infection for mimicking arbovirus transmission and has been a successful route for evaluating Chikungunya virus infection in NHPs (Crimotich et al. Virology Journal, 2017). Future studies are needed in NHP models to evaluate the ID route of ZIKV infection. To address this, we modified our statement as follows on Lines 417-419: “The ZIKV inoculum was based on previous NHP studies and the subcutaneous route of administration over several sites was implemented to mimic the natural infection route of a probing mosquito and this administration had been successful in NHPs”

4. “presence of antibodies to West Nile, Dengue, and Zika” Clear. I assume no risk of other flavivirus exposures either from natural circulation or past direct or indirect exposure in the context of research?

Response: As mentioned in the methods (Lines 423-425): “All animals were pre-screened for the presence of antibodies to West Nile, Dengue, and Zika viruses, and all were negative except for a single female animal (L07226), who was seropositive for West Nile.” These animals did not receive direct research exposure to any other flaviviruses prior to assignment to these studies. All animals were born and housed in the USA, however, at this time we cannot preclude that these animals were naturally exposed to other flaviviruses (excluding WNV, DENV, ZIKV) prior to assignment.

5. “ketamine 10 mg/kg intramuscular” Is there literature on impact of anesthesia on immune responses such as cytokine and chemokine secretion?

Responses: A study by Lugo-Roman et. al (J Med Primatol, 2009) demonstrated that ketamine administration had the potential to modify hematology and serum biochemistry, therefore ketamine may have an impact on cellular immune responses including cytokine and chemokine secretion. Therefore, we added the following statement to the Discussion to address this possibility. This also addresses the concern raised above for Methods, Point 1 and a Major Concern (Point 6) from Reviewer 1 (Lines 355-358): “In addition, other factors such age, weight, and repeated ketamine sedation could alternatively influence cellular immune responses and associated sex differences” and Lines 455-458: “Macaques used on infectious disease studies are nearly universally sedated with ketamine for procedures to ensure staff safety and permit collection with minimal stress to the animals. It is therefore also the best choice to permit comparison to the other literature generated on ZIKV in macaques, which also used repeated ketamine sedation during their studies (Dudley et al, Nat Communications, 2016; Hirsch et al, PLOS Pathogens, 2017; Coffey et al, PLOS One, 2017).”

REVIEWERS' COMMENTS:

Reviewer #1 (Remarks to the Author):

The revised manuscript is acceptable.

However the response to query #6 should include the presentation of pre-infection values (baseline). The AUC values presented do not take into consideration the quite variable baseline data.

REVIEWERS' COMMENTS:

Reviewer #1

The revised manuscript is acceptable. However the response to query #6 should include the presentation of pre-infection values (baseline). The AUC values presented do not take into consideration the quite variable baseline data.

*Response: We agree with the reviewer that inclusion of pre-infection (baseline) values in the AUC immune analysis is important for the interpretation of the immune responses. For this reason, our initial AUC analysis included these values in the calculation, however we were not clear in our explanation of this analysis in the text and figures. Inadvertently we referred to baseline/pre-infection timepoints as D0, rather than D-10, therefore it was unclear that pre-infection values were taken into consideration. In response, we have clarified the method and rationale for using AUC analysis in the text, Lines 179-190: “To account for differences in pre-infection immune frequencies between the sexes that could influence the kinetics and magnitude of the host response to ZIKV infection, we evaluated the area under the curve (AUC) of the immune cell responses. AUC analysis was between days -10 to +4 in the blood and between days -10 to +21 in the tissues.” In addition, we have changed all the necessary X- and Y-axis labels on all AUC figures in **Figure 2 and 5** and the Figure/Table legends (**Figures 5, 2; New Supplementary Table 3**) to clarify this methodology. In particular we have added the following to the text in the results section to address differences in baseline levels of neutrophils in the blood between males and females, Lines 251-253: “Initial baseline levels of neutrophils in the blood and mucosal tissues were also higher in the females which may be a factor that contributed to greater recruitment of these cells in the females after ZIKV infection.”*

Figure 2. Innate immune cells are rapidly recruited in response to ZIKV infection.

Frequencies within $CD45^+$ leukocytes and area under the curve (AUC) analysis of (a) pDCs ($CD20^+CD3^-HLA^-DR^+CD14^-CD123^+CD11c^-$), (b) mDCs ($CD20^+CD3^-HLA^-DR^+CD14^-CD123^-CD11c^+$), (c) monocytes ($CD20^-CD3^-HLA^-DR^+CD16^+CD14^+$), and (d) neutrophils ($CD3^+CD11b^+CD14^+HLA-DR^-SSC-A^{Hi}$), in whole blood, peripheral lymph node (PLN), and rectum throughout the course of ZIKV infection were measured by flow cytometry. The frequency of these responses in the blood AUC analysis in blood (AUC, days -10 to +4) and tissues (AUC, days -10 to +21) are shown. (a-d) Blue (females) and red (males) symbols in graphs represent individual animals, with curves indicating the median response for all animals over time.

Comparisons of responses at each timepoint versus baseline were conducted by paired Wilcoxon test. Unadjusted p -values ≤ 0.05 are displayed, all p -values are available in Supplementary Table 2. (a-d) Differences in AUC between males and females were also determined using the Wilcoxon test, with significant unadjusted p -values ($p \leq 0.05$) shown. Dots represent individual animals (total $n=8$), female (blue) and male (red), with bars indicating median with interquartile ranges.

Figure 5. Innate responses in blood recruit viral targets to tissues and promote viral persistence. a) Early cellular innate immune responses in PTM with and without detectable ZIKV RNA in the rectum at day 7. Each panel shows the AUC of pDCs measured by flow cytometry in the blood between days -10 to +4 or the concentration of plasma MCP-1 measured at 2 dpi in animals with or without detectable ZIKV in the rectum at 7 dpi. Lines indicate the median. Differences between animals with no virus or virus in the rectum were evaluated using Wilcoxon tests. Unadjusted p-values are shown. (b) Correlations between viral burden in the PLN (AUC, days -10 to +21) and either pDC frequency (AUC, days -10 to +4) (left panel) or the concentration of plasma MCP-1 measured at 2 dpi (right panel). (c) Scatter plot and Spearman correlation analysis of the relationship between the concentration of plasma MCP-1 measured at 2 dpi and mDCs (AUC, days -10 to +21) and non-classical monocytes (AUC, days -10 to +21) in the PLN. Significance of the unadjusted Spearman correlation test p-values are indicated as follows: * $p \leq 0.05$, ** $p \leq 0.01$, or ns (not significant).

Cell Subset	Blood	PLN	Rectum	Jejunum	Colon
pDCs	0.0286	0.8857	0.3429	0.2000	0.0286
mDCs	0.2000	0.1143	0.3429	0.2000	0.3429
Monocytes	0.6857	0.4857	0.0286	0.0286	0.0286
Classical	0.4857	0.1143	0.0286	0.0286	0.0286
Intermediate	0.8857	0.3429	0.0286	0.8857	0.0286
Non-Classical	0.4857	0.0286	0.0571	0.0286	0.0286
Neutrophils	0.0286	0.3429	0.0286	0.0286	0.0286
NK Cells	0.4857	0.2000	0.4857	0.3429	0.3429

Supplementary Table 3. Comparison of cellular innate responses between males and females in different tissues. Unadjusted Wilcoxon test p-values comparing AUC of cell frequencies of CD45⁺ leukocytes in blood (days -10 to +4) and tissues (days -10 to +21) between males and females, with $p \leq 0.05$ indicating significant differences (italicized and bolded) and $p \leq 0.08$ indicating trending differences (italicized). Colors indicate the direction of difference, i.e. higher in females (blue) or males (red).